# Projections of adult skills and the effect of COVID-19

Caner Özdemir[1]*, Claudia Reiter[2,3], Dilek Yildiz[3,4], Anne Goujon[3,4]

**1** Department of Labour Economics and Industrial Relations, Zonguldak Bülent Ecevit University, Zonguldak, Turkey, **2** Department of Demography, University of Vienna, Vienna, Austria, **3** Population and Just Societies Program, International Institute for Applied Systems Analysis, Laxenburg, Austria, **4** Vienna Institute of Demography, Austrian Academy of Sciences, Vienna, Austria

* canerozdemir@beun.edu.tr

**Citation:** Özdemir C, Reiter C, Yildiz D, Goujon A (2022) Projections of adult skills and the effect of COVID-19. PLoS ONE 17(11): e0277113. https://doi.org/10.1371/journal.pone.0277113

**Data Availability Statement:** Data and codes used to generate the results are available in the GitHub repository: https://github.com/ozdemircaner/SLAMYS-projections

**Funding:** The author(s) received no specific funding for this work

## Abstract

In this paper, we project Skills in Literacy Adjusted Mean Years of Schooling (SLAMYS) for the working age population in 45 countries and quinquennial time periods until 2050 according to various population scenarios. Moreover, we integrate the effect of school closures due to the COVID-19 pandemic on these projections. Adult skills are projected using the cohort components method. They can help in assessing the potential consequences of the recent trends for the adult population, particularly the workforce, whose skills are essential for the jobs contributing to economic growth and development outlooks. Our projections are novel as they take into account both the amount of schooling and quality of education and also consider the changes in adult skills through lifetime. Projections show that the adult skills gap between countries in the Global North and countries in the Global South will likely continue to exist by 2050, even under very optimistic assumptions–but may widen or narrow depending on the demographic development trajectories specific to each country. Moreover, the loss of learning due to school closures during the COVID-19 pandemic further exacerbates inequalities between countries. Particularly, in countries where schools have been closed for a prolonged period of time and the infrastructure for effective online schooling is lacking, the skills of cohorts who were in school during the pandemic have been severely affected. The fact that the duration of school closures has been longer in many low- and middle-income countries is a serious concern for achieving global human capital equality. The impact of the COVID-19 pandemic is projected to erase decades-long gains in adult skills for affected cohorts unless policies to mitigate learning loss are implemented immediately.

## Introduction

Quality education for all is a sustainable development goal (SDG) to the horizon 2030. Set in 2015, SDG4 is "to ensure inclusive and equitable quality education and promote lifelong opportunities for all" [1]. This means wider acceptance of the need for the improvement of quality of education along with educational expansion. However, while monitoring progress

**Competing interests:** The authors have declared that no competing interests exist.

of attainment such as completion rates of primary and lower secondary is feasible, controlling for the quality of education is more difficult. There is evidence that more children and youth have access to education over the years, but some indicators point at a decline in the outcome of that education in terms of acquired skills (whether it is advanced literacy or numeracy or else) [2–5]. When those skills are not acquired during schooling, their lack will most likely accompany individuals throughout their lifetime in the absence of compensatory adult education.

Projecting the education and the skills acquired using the cohort component method, which is the main aim of this paper, can help in assessing the potential consequences of the recent trends for the adult population, particularly the workforce, whose skills are essential for the jobs contributing to economic growth and development outlooks. Furthermore, developing adult skills through quality education is likely to not only boost economic progress but also support social mobility towards more productive and more rewarding jobs [6]. Thus, projecting the skills of the adult workforce would give an insight into the future status of societies. The COVID-19 pandemic has added a challenge to improve the quality of education by forcing many countries to close schools for several weeks with potential long-term consequences on skills learned during school time.

In this sense, this paper aims at projecting adult skills until 2050 and measuring the effect of school closures due to the COVID-19 pandemic on these skills. The future of skills and skill loss due to the pandemic-induced school closure is quantified by projecting SLAMYS until 2050. SLAMYS is a novel indicator for human capital, recently introduced by Lutz et al. [2], which aims at capturing the quantitative and qualitative dimensions of education and learning. The quantitative dimension is measured by mean years of schooling (MYS) whereas the qualitative dimension is measured by a so-called Skills-Adjustment Factor (SAF) that is based on adult literacy test scores. Lutz et al. [2] provided SLAMYS estimates for the 20-64-year-old population in 185 countries for quinquennial time periods from 1970 to 2015.

In this paper, we project SLAMYS until 2050 for 45 countries by incorporating projections for both the average length of schooling and the quality of education. Recent research showed that focusing only on the educational degrees or length of schooling is not enough since schooling does not necessarily mean learning [3]. Although many developing countries have expanded basic education and have increased the educational attainment level of their population over the last decades, there are still huge gaps (especially in terms of quality of education) between the Global South and the Global North [7–9]. This can be also tracked in SLAMYS estimates between 1970 and 2015. Despite converging MYS averages between developed and developing countries, the difference between average literacy skills has been increasing [2].

Since human capital theory has become widely accepted in the social sciences, scholars have primarily used the amount of schooling as an indicator of human capital in their analysis–whether it is the completed level of education (e.g., lower secondary education) or the number of years of schooling [10–12]. Although this approach has been widely criticized [13–15], limitations in data availability have prevented researchers from using alternative measures. As international student performance assessments have become available, it has been shown that students in different countries at similar stages of their education do not have the same competencies [4,16–18]. The availability of these data allowed scholars to use student assessment scores as a proxy for the quality of education and human capital [19–21]. Nevertheless, their approach has also its flaws since it uses the performance of the school-age population as a proxy for the whole working-age population. SLAMYS is able to rectify both the issue of focusing only on quantity dimensions of education (by considering both years of schooling and skills), and the inconsistency of using data collected from children to represent the working-age population (by using adult skills assessments).

Another major contribution of our work is that we consider the effect of school closures during the COVID-19 pandemic. Starting from March 2020, many countries adopted measures to prevent the spread of the virus including school closures, travel restrictions and imposing home-office. In many cases, schools remained closed for weeks if not for months. Although most countries switched to online teaching, there were large inequalities between and within countries [22–25]. Since the very start of the pandemic, researchers have tried to quantify the effect of school closures on learning. In the early days of the COVID-19 pandemic, researchers attempted to estimate potential learning losses due to school closures based on existing evidence after natural disasters and summer breaks [26–29]. In one of these, Kaffenberger [28] argued that three months of school closure may result in a learning loss that amounts to more than one year of learning gains, but she also claimed that most of these losses can be recovered through remediation programmes after returning to school. Some researchers used educational data sources on skills and learning that were collected before and after the pandemic [24,25,30–35]. For example, Engzell and others [25] used a national exam data in the Netherlands, which was conducted just before and after an eight-week-school lockdown. Although this was a relatively short break and the Netherlands were more prepared to online teaching than many other countries around the world, the study found a learning loss of about 3% (or 0.08 standard deviations) in maths, spelling, and reading compared to previous years. There are also studies that use data which were collected during the pandemic [36–39]. For example, comparing PIRLS 2016 and 2021 data, Ludewig et al. [38] found learning loss in reading scores among fourth graders in Germany to be equal to one-third of a year of learning.

Recently, some researchers published meta-reviews on the amount of learning losses during the pandemic [40–43]. All reviews documented learning losses around 0.1 to 0.15 standard deviations across different countries, grades and subjects. They also report that these losses are not homogenous across populations, with children from disadvantaged backgrounds having been the most affected.

In order to understand the potential amount of learning losses in different settings, we use the simulations done by Azevedo et al. [29]. They ran simulations of learning losses due to school lockdowns during the pandemic using data from 174 countries and estimated learning losses according to different length of school closures and by country income groups, taking into account the remediation potentials of countries of varying development levels. Building on this research [29], we project the effects of learning loss during the pandemic on future adult skills.

## Data and methods

As emphasized above, SLAMYS has two dimensions: a quantitative and a qualitative one. The quantitative dimension is measured by MYS, whereas the qualitative dimension is measured by the SAF. SLAMYS is the product of these two dimensions:

$$\text{SLAMYS} = \text{MYS x SAF} \tag{1}$$

Consequently, both dimensions need to be projected. For the quantitative dimension, we used the existing projections from the Wittgenstein Centre (WIC) Human Capital Data Explorer [44]. Here, MYS is projected for quinquennial time periods for virtually all countries around the globe, according to alternative demographic scenarios called Shared Socioeconomic Pathways (SSPs) [45]. Among them, we used three major scenarios. The first one is SSP1, the rapid development scenario. In this scenario, it is assumed that all countries reach the education SDG by 2030 and achieve low mortality and fertility rates at fast pace. The second scenario, SSP2, is the baseline scenario that assumes a continuation of the trends in the

future, with countries achieving medium mortality, fertility and educational expansion. The third scenario, SSP3, is the stalled development scenario. In this scenario, global social development is slowing down, and inequalities are increasing. This is associated with higher mortality and fertility, and slower educational expansion. MYS projections for the three scenarios are used in the SLAMYS projections.

The SAF, which represents the qualitative dimension of SLAMYS, is based on literacy scores in adult skills tests such as those measured by the Programme for the International Assessment of Adult Competencies (PIAAC) [5], the Skills Measurement Programme (STEP) [46] and the International Adult Literacy Survey (IALS) [47]. The SAF is assumed to be equal to 1 for the OECD average literacy score in every education-age group. Thus, if a particular education-age group in a country has average literacy skills higher than its peers in OECD, its SLAMYS will be higher than its MYS, if this particular group score lower in the test than its peers in OECD, it will have a lower SLAMYS compared to its MYS. The SAF could only be calculated for the 45 countries that recently participated in one of the above-mentioned international adult skills tests. Although Lutz et al. [2] estimated the SAF for the remaining countries by using various socioeconomic indicators as regressors, such indicators are for the most part not available for future time periods.

In order to project the SAF to the future, three steps are employed: (1) current cohorts are aged; (2) trends in student performance are applied onto emerging cohorts for different SSP scenarios; and (3) the effects of learning loss are applied for cohorts aged between 6 and 15 who were in school during the pandemic.

As for the first step, birth cohorts in 2015 are aged according to the age- and education- specific changes in adult skills [2,48]. Reiter [48] showed that adult literacy skills do not remain the same across one's lifetime and these changes vary significantly depending on the level of education. Higher educated people continue building up their skills until the age of 30–34, after that their skills remain constant for some time and eventually start decreasing after the age of 45. On the other hand, the literacy skills of less educated individuals start declining right after they leave schools. For the projection of the SAF into the future, cohorts in 2015 are aged until 2050 using these education-specific adult skills changes. In addition, to be able to apply these changes to emerging cohorts, we also need to project the starting SAF value at the age of 15. To project the literacy skills of 15–19-year-olds in the future, PISA (Programme for International Student Assessment) reading performance of 15-year-old students are used in combination with three SSP scenarios. PISA has been conducted every three years since 2000 for more than 80 countries and measures reading skills of 15-year-old students in a very similar fashion as PIAAC [4]. Using reading performance data from 7 waves of PISA, five-year-average changes in reading scores are calculated for each participating country, which range between ±3%. For the baseline scenario, SSP2, the ongoing five-year trends in countries' PISA reading scores of countries are assumed to continue until 2030. More precisely, if the PISA reading performance of 15-year-old students in a country has been increasing by 1% on average every five years, it is assumed that the SAF of 15–19-year-olds in this country would also increase by 1% in the 2020–24 period compared to 15–19-year-olds in 2015–19. While it is only possible to calculate country-specific trends for countries that have participated in at least two waves of PISA, trends in countries with similar development levels are used for the countries who have not participated in PISA more than once. After the 2030–34 period, all countries are assumed to follow a 1.5% increase (average five-year increase of countries that improved their PISA reading scores between PISA waves) in the SAF of 15-year-olds until 2050. For the SSP1 scenario, all countries are assumed to increase their SAF by 3% (highest five-year-average increase trend in PISA among 45 countries: Chile and Peru) from 2015 to 2050. Finally, for the SSP3 scenario, it is assumed that emerging cohorts have the same SAF as the previous cohort, i.e., no improvement in quality. Fig 1 shows the SAF

of 15-19-year-olds from 2015 to 2050 across SSPs in three selected countries with differing PISA trends: USA (stable), Singapore (increasing), the Netherlands (decreasing). Then, for all scenarios, these emerging cohorts are aged according to the age- and education-specific changes in adult skills as explained above. In this way, the SAF can be estimated for every five-year age group and every five-year period from 2015 to 2050 in 45 countries.

Finally, the potential effects of learning loss during the COVID-19 pandemic are also incorporated to the SAF of cohorts that have been in school during the pandemic. As underlined above, it is hard to dismiss the potential effects of months of school closures on learning and thus on future adult skills of affected cohorts. Based on simulations, Azevedo et al. [29] have estimated the changes in Learning Adjusted Years of Schooling (LAYS) in countries with varying income levels and school closure lengths (See Table 1).

The percent change in LAYS is assumed to be the same as the change in the SAF, since LAYS are calculated in a very similar manner as SLAMYS. Indeed, LAYS is an indicator measuring the quality of education by adjusting MYS just like SLAMYS do [49]. However, instead of adjusting with adult skills as in SLAMYS, in LAYS, MYS is combined with a learning score calculated using international student assessment data. Therefore, depending on the World Bank income classification and the length of school closure in the respective country, the SAF is adjusted for the cohorts in school during the pandemic, i.e., those born between 2005–2014.

Data for the length of school closures are taken from UNESCO [50]. Since the early weeks of the pandemic, UNESCO has classified daily status of schools in almost every country as open, closed due to the COVID-19, partly closed due to the COVID-19, and academic break. We converted the daily data to weekly data assuming 7 days of closure or 14 days of partial closure as one week of school closure, then countries are assigned to the school closure scenarios of Azevedo et al. [29] (Fig 2).

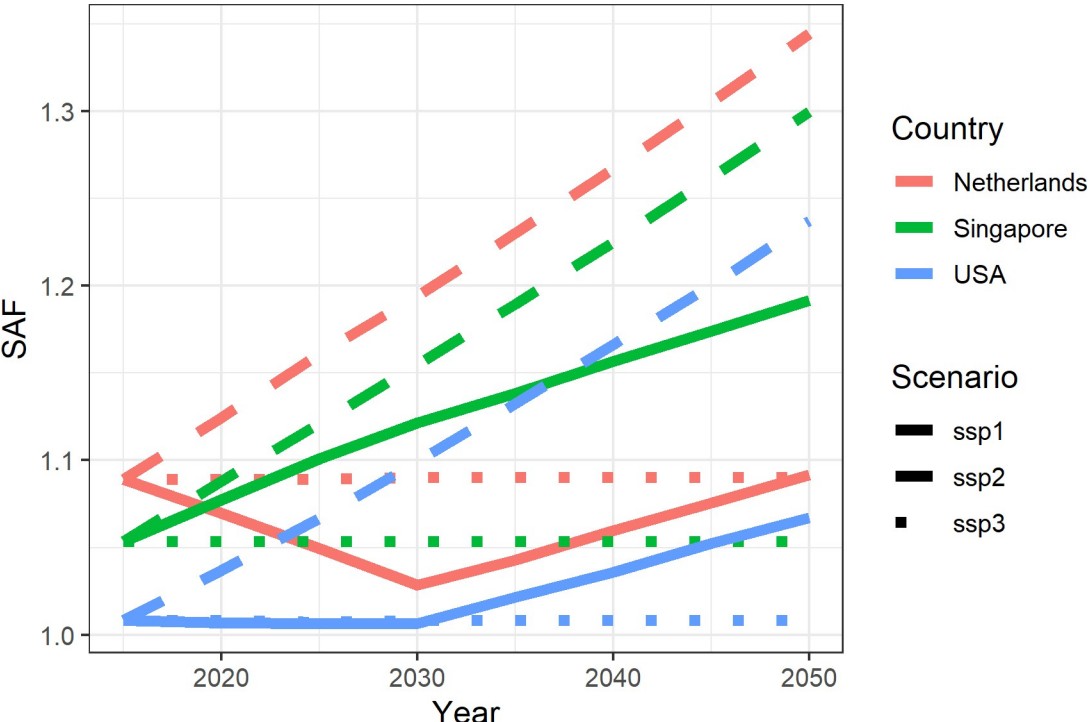

**Fig 1. Projection of SAF of 15–19 age group, several scenarios, 2015–2050.** Source: Authors' own calculations.

**Table 1. The effect of school closures on Learning-Adjusted Years of Schooling (LAYS).**

| Income Level | | School closure scenarios | | | |
|---|---|---|---|---|---|
| | Baseline | Optimistic | Intermediate | Pessimistic | Very pessimistic |
| High | 10.3 | 10.0 | 9.6 | 9.2 | 8.9 |
| Upper middle | 7.8 | 7.5 | 7.2 | 6.9 | 6.7 |
| Lower middle | 6.6 | 6.3 | 6.0 | 5.8 | 5.6 |
| Low | 4.3 | 4.1 | 3.9 | 3.8 | 3.6 |

Note: Adapted from Azevedo et al. [29].

## Results

Following the steps and scenarios explained above, four alternative SLAMYS projections are calculated for five-year time periods until 2050 for the working-age population (20–64) in 45 countries. Among these scenarios, the first one follows the SSP2 scenario without the pandemic learning loss effects. This scenario is calculated to build a benchmark for the other three. The other scenarios are SSP1, SSP2 and SSP3 with pandemic learning loss effects.

Fig 3 shows MYS and SLAMYS projections for all four scenarios for the year 2050. WIC projections for MYS (in grey bars) show that many countries of the Global South may catch up with the Global North or at least narrow the schooling gaps particularly for the SSP1 and the SSP2 scenarios. However, SLAMYS projections reveal a more pessimistic image in this sense. Regarding the benchmark scenario, SSP2 without the effect of COVID-19 learning losses, many countries are still expected to have lower SLAMYS than their MYS in 2050. Particularly for the two African countries in our sample, Ghana and Kenya, the difference is more than 7 years.

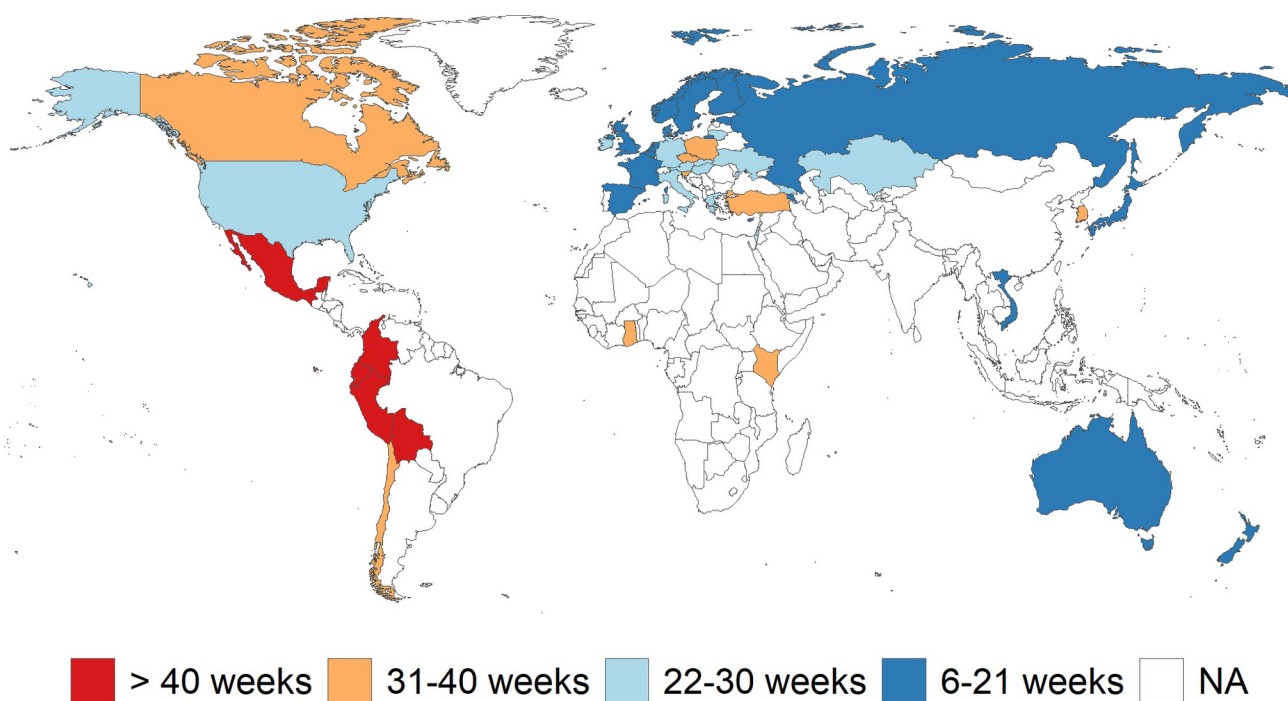

**Fig 2. Length of school closures.** Note: Base map was constructed using the R packages rnaturalearth and sf with free vector and raster map data at https://naturalearthdata.com. Data come from the UNESCO school closure data at https://en.unesco.org/covid19/educationresponse.

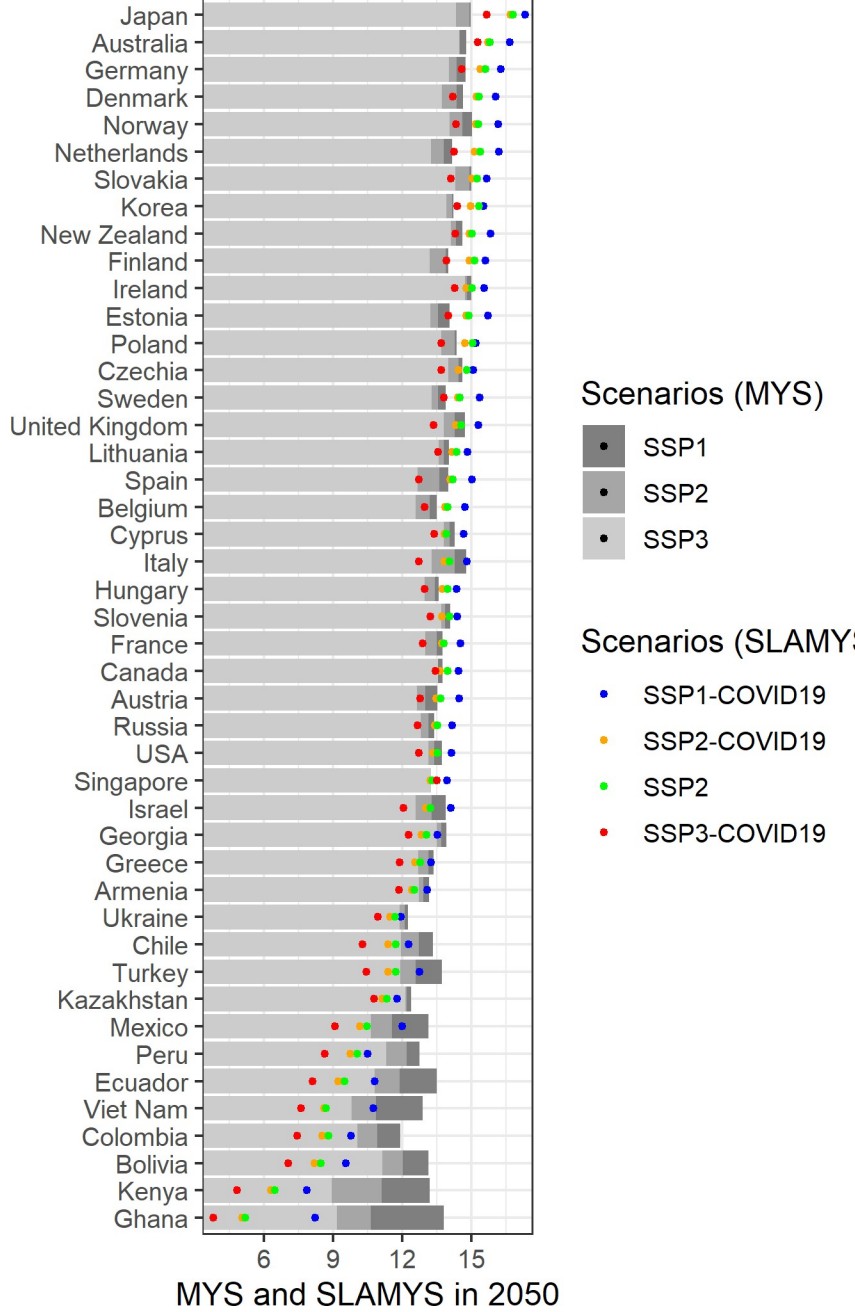

**Fig 3. Projections of MYS and SLAMYS, several scenarios, 2050.** Source: Authors' own calculations. Note: Data sorted by projected SLAMYS according to SSP2-COVID19 scenario.

Another striking fact in Fig 3 is that the differences between scenarios (for both MYS and SLAMYS) are larger in the Global South countries of the sample while there is less variation among the Global North countries. Moreover, SLAMYS differences remain high even in the best-case scenario SSP1 –although the gaps between countries in terms of mean years of schooling are narrowing.

To show the effect of learning loss, Fig 4 focuses on the cohort of the 15–19-year-olds who were in school during the pandemic and shows the SLAMYS outcomes for 40-44-year-olds–the

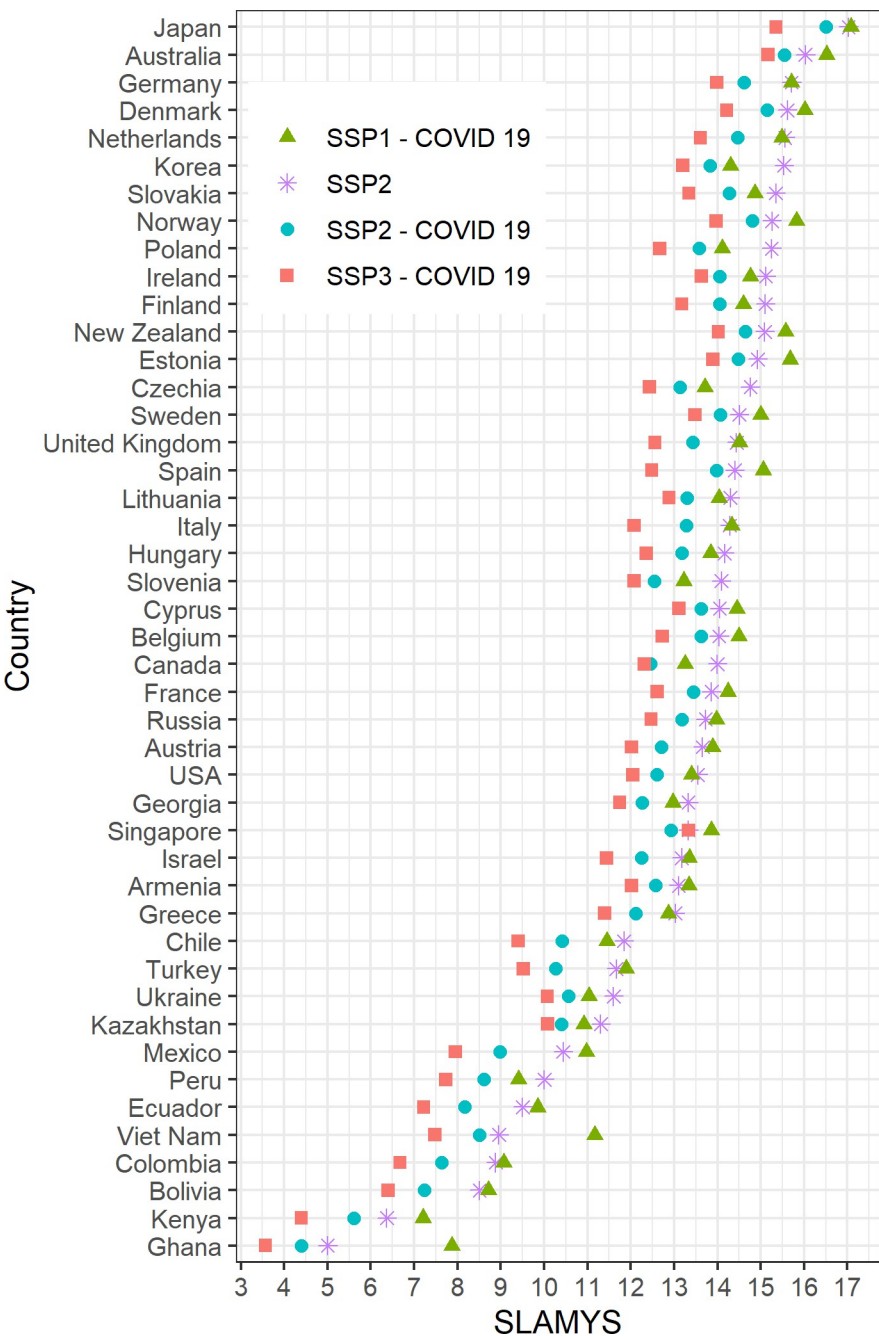

**Fig 4. Projection of SLAMYS, 40–44 year-olds, several scenarios, 2050.** Source: Authors' calculations. Note: Data sorted by projected SLAMYS according to SSP2 scenario.

age when SLAMYS is at its peak for highly educated population [51]–in 2050, according to four scenarios. The SSP2 scenario without the pandemic learning loss effects is also included in Fig 4 in order to enable comparisons. For countries that have experienced shorter school closures, the gap between SSP2 scenarios with and without pandemic learning loss effects is smaller. On the other hand, in countries such as Mexico, Ecuador, Peru, Colombia, Poland, Korea, and Turkey, where the school closures were longer the differences amount for more than 1.5 years.

To have an even clearer picture of the effects of learning loss during the pandemic, Fig 5 shows SLAMYS projections using the baseline scenario SSP2, with or without school closures for 40-44-year-olds in 10 selected countries (Figure for all countries are provided in SI section). There are two projections available for the cohorts born between 2005–2009 and 2010–2014: with and without learning loss effects due to pandemic school closures. In the upper row, there are five countries where schools were closed for a relatively short time, both projections with and without the pandemic learning losses are very similar. However, for countries in the lower row where face-to-face teaching was suspended for a longer period, the losses are much bigger. For example, countries such as Peru, Poland or Turkey, which have recently experienced a rapid expansion in schooling and an increase in student performance in international exams, have been much more affected by the pandemic due to longer periods of school closure. Despite rapid progress in the recent past, in these cases, SLAMYS is projected to drop to the levels of cohorts born 15 to 20 years earlier when the pandemic learning loss is taken into consideration.

## Discussion & conclusion

In this paper, we extend the research on combining school attainment and acquired skills by projecting SLAMYS (2) for 45 countries up to 2050. Estimates of SLAMYS for the past have revealed a widening global skills gap between low and high performing countries, with significant impact on inequalities in socio-economic development between populations [2]. While that alone is cause for concern, the on-going COVID-19 pandemic caused significant disruption to the global education system, putting an additional burden on societies. In our projections, we therefore explicitly consider the potential impact of the COVID-19-related school closures on adult skills. Based on recent simulations of the influence of the pandemic on schooling and learning, trend data from both international student and adult assessments as well as projections of educational attainment following the Shared Socioeconomic Pathways,

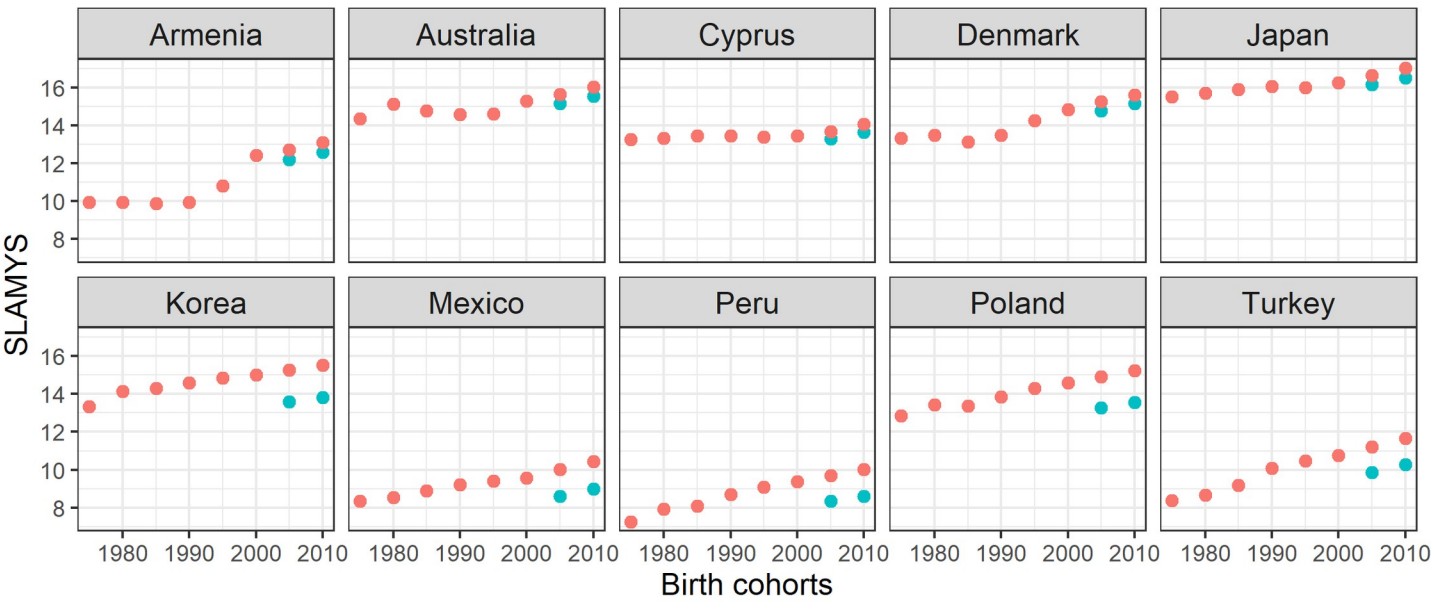

**Fig 5. Projected SLAMYS (SSP2) for 40–44 year-olds across birth cohorts.** Source: Authors' own calculations.

we have developed a set of four scenarios (three scenarios with the effect of learning losses during the COVID-19 pandemic and one benchmark scenario without this effect) for future human capital in 45 countries around the world.

Our results show that the skills gap between high-income and low-income countries will likely continue to exist by 2050 even under very optimistic assumptions–but may widen or narrow depending on the demographic development trajectories specific to each country. If poorer countries manage to expand schooling and improve skills to the levels set by the SDGs by 2030 and experience lower mortality and fertility as projected in the SSP1 scenario, the younger cohorts of these countries could achieve SLAMYS similar to those of their peers in richer countries. On the contrary, under the pessimistic SSP3 scenario, gaps in terms of adult skills are projected to further increase. As a consequence of the interaction between high fertility and lower educational attainment [52], the number of children born to low educated mothers will increase and the share of population with little or no education will grow–resulting not only in lower literacy skills to start with, but also in more people that will start losing their skills earlier in life. Our estimates of the increase in the global human capital gap with the exception of the optimistic scenario are consistent with earlier studies using quantity of schooling, i.e. educational attainment, as a proxy for human capital [45,53]. Previous evidence has shown that human capital is strongly correlated with development [53–55]. Results of international student assessments such as PISA, TIMSS and PIRLS or adult skills tests like PIAAC or STEP illustrated that there are already huge human capital gaps between countries and they have not been narrowing down for the last decades [4,5,16,17,46]. Using SLAMYS, our projections show the translation of these patterns in the future when no correction measures or pessimistic scenarios are being followed. Since the differences in SLAMYS between scenarios are the largest for countries in the Global South, these countries stand at a crossroads, with their future largely depending on the continued expansion of school enrolment and quality of education.

Moreover, learning loss due to school closures during the COVID-19 pandemic further exacerbates the inequalities between countries. Particularly, in countries where schools have been closed for a prolonged period of time and the infrastructure for effective online schooling is lacking, the skills of cohorts who were in school during the pandemic have been severely affected. The fact that the duration of school closures has been longer in many low- and middle-income countries is a serious concern for achieving global human capital equality. There have been plenty of research investigating the learning loss during or just after the school closures due to the pandemic. Almost all of them found significant learning losses [24,25,27,31–36,38,39]. Early meta-reviews showed that virtually all studies found learning losses in different contexts [22,41–43,56]. Though, most of the studies indicate that these losses vary across countries or regions within a country. Especially pupils from disadvantaged socio-economic groups are found to be experiencing more severe learning losses. Moreover, some studies indicated that these losses may result in reduction in economic growth in the future [57]. The impacts of learning losses in the aftermath of COVID-19 are in line with that of other past events causing school disruptions such as wars (for instance [58] or natural disasters [59]. Our projections also indicate a worrying picture. The impact of the COVID-19 pandemic is projected to erase decades-long gains in adult skills for affected cohorts unless policies to mitigate learning loss are implemented immediately. This could seriously compromise the achievement of SDG 4 in many countries and therefore require further efforts than those already needed to progress successfully toward the goal [60].

The main research contribution of this paper can be summarized as follows. On the one hand, we provide for the first time estimates of future human capital capturing not only the quantitative dimension (i.e., the educational attainment) but also the qualitative dimension (i.e., the actual skills) of human capital, with clear relevance for progress towards development goals. By projecting SLAMYS for several birth cohorts until 2050, the future of adult human

capital is assessed for a set of scenarios and countries. On the other hand, we quantify the mark that the COVID-19 pandemic is likely to leave on the skills of countries with varying levels of development around the world. By focusing not only on the immediate learning loss due to school closures but also on the potential long-term effects of the COVID-19 pandemic on the human capital in the coming decades, our results stress the urgency of introducing mitigation measures and recovery strategies in response to the tremendous disruption to the global education system.

As with the majority of studies, the design of the current article is subject to some limitations. First and foremost, it covers a very specific set of countries, most of which are wealthy OECD countries in the Global North. In order to be able to increase geographic coverage and extend our analyses (and conclusions drawn from them) to a wider range of countries, large-scale adult skills assessment data would be needed for more low- and middle-income countries. Second, the skills dimension of SLAMYS comprises a very specific domain of skills, namely literacy skills, and rests on the implicit assumption that they can be reliably assessed through tests. Despite studies having shown that literacy skills are closely correlated with other type of skills, one should be cautious when transferring these results to all kinds of competencies. Furthermore, it is important to stress that–as with any projections–our results strongly depend on the underlying assumptions of each scenario. The projections that rely on what-if scenarios are not predictions. Rather, our aim is to explore the consequences of different demographic and skills trajectories on the future of human capital. Also, the exact impact of the COVID-19 pandemic remains largely uncertain, given that the pandemic is still on-going, and new information and data are published on a rolling basis. Only time will tell whether students quickly rebound from the COVID-19-pandemic related learning loss, or whether the setback will have long-lasting consequences for this generation, especially when additional health issues, in particular related to mental health, become more prevalent as a direct and indirect impact of the pandemic [61]. The range of estimates presented in this paper are therefore also subject to the uncertainty inherent in the situation. Lastly, the data we use in this study are at the country level and do not take into account differences within country. In addition to an immense literature on inequalities in the quantity and quality of education, recent studies provide vast evidence on within-country inequalities in learning loss during the COVID-19 pandemic and access opportunities to online learning. Thus, future research may also focus on intranational inequalities.

## Supporting information

**S1 Fig. Projected SLAMYS (SSP2) for 40-year-olds across birth cohorts.** Source: Authors' own calculations.
(TIF)

## Acknowledgments

We would like to thank Prof. Wolfgang Lutz for his valuable feedback during various stages of the research. We also would like to thank participants in the 2021 International Population Conference and Population Association of America 2022 Annual Meeting for their constructive and useful comments.

## Author Contributions

**Conceptualization:** Caner Özdemir, Claudia Reiter, Dilek Yildiz, Anne Goujon.

**Data curation:** Caner Özdemir, Claudia Reiter, Dilek Yildiz.

**Formal analysis:** Caner Özdemir, Claudia Reiter, Dilek Yildiz.

**Methodology:** Caner Özdemir, Claudia Reiter, Dilek Yildiz, Anne Goujon.

**Visualization:** Caner Özdemir, Claudia Reiter, Dilek Yildiz, Anne Goujon.

**Writing – original draft:** Caner Özdemir, Claudia Reiter, Dilek Yildiz, Anne Goujon.

**Writing – review & editing:** Caner Özdemir, Claudia Reiter, Dilek Yildiz, Anne Goujon.

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
