## [Decision Letter · Decision Letter 0]

19 Sep 2022

PONE-D-22-19138Projections of adult skills and the effect of COVID-19PLOS ONE

Dear Dr. Özdemir,

Thank you for submitting your manuscript to PLOS ONE. After careful consideration, we feel that it has merit but does not fully meet PLOS ONE’s publication criteria as it currently stands. Therefore, we invite you to submit a revised version of the manuscript that addresses the points raised during the review process.

We look forward to receiving your revised manuscript.

Kind regards,

Petri Böckerman

Academic Editor

PLOS ONE

Journal Requirements:

2. We note that Figure 2 in your submission contain [map/satellite] images which may be copyrighted. All PLOS content is published under the Creative Commons Attribution License (CC BY 4.0), which means that the manuscript, images, and Supporting Information files will be freely available online, and any third party is permitted to access, download, copy, distribute, and use these materials in any way, even commercially, with proper attribution. For these reasons, we cannot publish previously copyrighted maps or satellite images created using proprietary data, such as Google software (Google Maps, Street View, and Earth). For more information, see our copyright guidelines: http://journals.plos.org/plosone/s/licenses-and-copyright.

Additional Editor Comments:

The revised version should address all comments.

Reviewers' comments:

Reviewer's Responses to Questions

**Comments to the Author**

1. Is the manuscript technically sound, and do the data support the conclusions?

Reviewer #1: Yes

Reviewer #2: Yes

Reviewer #3: Yes

2. Has the statistical analysis been performed appropriately and rigorously? 

Reviewer #1: Yes

Reviewer #2: Yes

Reviewer #3: Yes

3. Have the authors made all data underlying the findings in their manuscript fully available?

Reviewer #1: Yes

Reviewer #2: Yes

Reviewer #3: Yes

4. Is the manuscript presented in an intelligible fashion and written in standard English?

Reviewer #1: Yes

Reviewer #2: Yes

Reviewer #3: Yes

5. Review Comments to the Author

Reviewer #1: The paper is timely, well written and addresses an issue that needs to be dealt with as the world goes into its post COVID19 pandemic phase. The methodology and the results obtained are clear and consistent. In my view the discussion of the results is somewhat deficient. In many cases the questions of "Why" and "What does this mean" are lacking. Oftentimes the discussion mirrors the way in which the results are written. I noted also that in spite of the fact that the paper has 53 references, the discussion is void of references except for lines 299 and 302 where reference 2 is used. Discussion section therefore needs to enhanced through a comparison with previous studies. While this is the first time we have had education affected in this by a COVID pandemic, it is not the first time that children have been denied an education due to factors such as disease (West Africa Ebola outbreak of 2013–2016), wars (world war 2) and natural disasters (Nearly 40 million children a year have their education interrupted by natural disasters). Publications in these scenarios can be used to enhance the discussion.

Reviewer #2: The manuscript is technically sound and well prepared. There is novelty in the topic and well supported with the literature review. Results are well presented though complex to understand. In the data and methods there is mention of three major scenarios for which the explanation is given. whereas in the results, there is a sentence "according to four scenarios" (line 268) which is not clear to the reader.

Reviewer #3: Well presented analysis on impact of school closure during COVID 19 on adult skills in due to impact of school closure,

The paper doesn't talk about any impact on how it may affect health or its correlation.

6. PLOS authors have the option to publish the peer review history of their article (what does this mean?). If published, this will include your full peer review and any attached files.

Reviewer #1: **Yes: **Devon Ronald Dublin

Reviewer #2: No

Reviewer #3: No

---

## [Author Response · Author response to Decision Letter 0]

13 Oct 2022

Dear Editor and Reviewers,

We are pleased to submit the revised version of our original research article entitled “Projections of adult skills and the effect of COVID-19” to be considered for publication in PlosOne. 

In this submission, we address the suggestions from the reviewers. First of all, we would like to thank for the constructive comments and suggestions. As per the editorial comment about Figure 2: The map used in the figure was constructed by the authors using R software. The base map is constructed using Natural Earth public domain maps, we added information about the R packages, cited Natural Earth web site and data used in drawing the map in the figure notes.

To meet the points raised by Reviewer #1, the Discussion & Conclusion section has been extended. A discussion about the increasing global skills gap is added in the second paragraph and relevant references were cited. Another discussion about previous evidence of learning loss was added in the third paragraph with the relevant citations.

Reviewer #2 pointed out the uncertainty about the number of projection scenarios used. It had been mentioned in the first paragraph of the Results section. Now, it is clearly stated once again in the first paragraph of the Discussion & Conclusion section.

In answer to the point of Reviewer #3 about the link to health, we have added a line in the Discussion & Conclusion section, reflecting on the potential long-term consequences of the pandemic on mental health. A detailed discussion would go, however, beyond the scope of this paper.

Thank you for your consideration,

The Authors

---

## [Decision Letter · Decision Letter 1]

20 Oct 2022

Projections of adult skills and the effect of COVID-19

PONE-D-22-19138R1

Dear Dr. Özdemir,

We’re pleased to inform you that your manuscript has been judged scientifically suitable for publication and will be formally accepted for publication once it meets all outstanding technical requirements.

Kind regards,

Petri Böckerman

Academic Editor

PLOS ONE

Additional Editor Comments (optional):

Reviewers' comments:

Reviewer's Responses to Questions

**Comments to the Author**

1. If the authors have adequately addressed your comments raised in a previous round of review and you feel that this manuscript is now acceptable for publication, you may indicate that here to bypass the “Comments to the Author” section, enter your conflict of interest statement in the “Confidential to Editor” section, and submit your "Accept" recommendation.

Reviewer #1: All comments have been addressed

Reviewer #2: All comments have been addressed

2. Is the manuscript technically sound, and do the data support the conclusions?

Reviewer #1: Yes

Reviewer #2: Yes

3. Has the statistical analysis been performed appropriately and rigorously? 

Reviewer #1: Yes

Reviewer #2: Yes

4. Have the authors made all data underlying the findings in their manuscript fully available?

Reviewer #1: Yes

Reviewer #2: Yes

5. Is the manuscript presented in an intelligible fashion and written in standard English?

Reviewer #1: Yes

Reviewer #2: Yes

6. Review Comments to the Author

Reviewer #1: (No Response)

Reviewer #2: Authors have addressed the reviewers' comments and done the modifications in the manuscript. Hence it can be accepted.

7. PLOS authors have the option to publish the peer review history of their article (what does this mean?). If published, this will include your full peer review and any attached files.

Reviewer #1: **Yes: **Devon Dublin

Reviewer #2: **Yes: **Prima Jenevive Jyothi D'Souza

---

## [Editor Report · Acceptance letter]

28 Oct 2022

PONE-D-22-19138R1 

Projections of adult skills and the effect of COVID-19 

Dear Dr. Özdemir:

I'm pleased to inform you that your manuscript has been deemed suitable for publication in PLOS ONE. Congratulations! Your manuscript is now with our production department. 

Kind regards, 

on behalf of

Professor Petri Böckerman 

Academic Editor

PLOS ONE